

# Individuals with fibromyalgia have a different gait pattern and a reduced walk functional capacity: a systematic review with meta-analysis

Elio Carrasco-Vega[1,2], María Ruiz-Muñoz[2,3], Antonio Cuesta-Vargas[1,2,4], Rita Pilar Romero-Galisteo[1,2] and Manuel González-Sánchez[1,2]

[1] Department of Physiotherapy, Faculty of Health Sciences, Universidad de Málaga, Malaga, Spain
[2] Institute of Biomedicine of Málaga (IBIMA), Malaga, Spain
[3] Department of Nursing and Podiatry, Faculty of Health Sciences,, Universidad de Málaga, Malaga, Spain
[4] School of Clinical Science, Faculty of Health Science,, Queensland University Technology, Brisbane, Australia

Corresponding author
María Ruiz-Muñoz, marumu@uma.es

## ABSTRACT

**Background**. The aim of the present study was to perform a systematic review and meta-analysis comparing walking test performance and gait pattern between individuals with and without fibromyalgia (FM).

**Methodology**. This systematic review was registered in PROSPERO with the following reference: CRD42018116200.The search for the scientific articles in this systematic review was carried out using the MEDLINE, SCOPUS, PEDRO, CINHAL and WEB OF SCIENCE databases. A combination of three conceptual groups of terms was used: (1) fibromyalgia; (2) walk (performance) tests; and (3) gait analysis. The included articles were analyzed for both functional and pattern of walking data of patients with FM. In order to provide a better estimate of the difference between individuals with and without FM on gait, a meta-analysis was performed on the 6MWT (6-minute walk test).

**Results**. Thirty-six studies were analyzed, with a total population of 4.078 participants (3.369 FM and 709 individuals without FM). From a functional point of view, the 6MWT distance covered by the group of individuals without FM was significantly greater than that of the individuals with FM in all the analyzed studies. In addition, when comparing the results obtained in the gait pattern analysis, it was observed that individuals with FM walked slower, with a shorter stride length and lower cadence compare to individuals without FM.

**Conclusions**. It is possible to affirm that individuals with FM perform walking tests differently than individuals without FM. It was observed that individuals with FM walk performing a cycle of shorter length and lower frequency, producing a slower gait, which results in a shorter distance traveled, in the same period of time, with respect to healthy subjects.

# INTRODUCTION

Fibromyalgia (FM) is a syndrome of unknown etiology (*Harris, 2008*), with a spectrum of symptoms that most commonly include the presence of generalized chronic pain, fatigue, sleep disturbance, mood disturbance and impaired quality of life (*Sumpton & Moulin, 2014*), causing a great impact on basic activities of daily living (BADL) (*Harris, 2008*). After osteoarthritis, FM is the second most common rheumatic disorder (*Clauw, 2014*), with 3–5% of the population suffering from FM (*Okifuji et al., 2016*), although, depending on the diagnostic criteria, the prevalence can reach 8% (*Clauw, 2014*). Therefore, it can become a great burden for patients, families and society in general. In Spain, it has been calculated that an individual with FM generates an average annual expense of over €5,000 compared to people without FM, of which 90% (*i.e.*, €4,500) are due to a loss of productivity (*Sicras et al., 2009*).

Although it is not an inflammatory process, it is accompanied by abnormal sensory responses (*Williams & Clauw, 2009*), usually expressed as hyperalgesia and/or allodynia (*Jon Russell, 1998*; *Carville & Choy, 2008*). Thus, a central sensitization mechanism has been described in FM (*Cuesta-Vargas et al., 2018*), causing an amplification of the neural signal in the central nervous system (*Yunus, 2000*), increasing the extension of the receptive field and unusually prolonging the pain even after the nociceptive stimulus disappears (*Mayer et al., 2011*). For this reason, one of the main goals of treating people with FM is to address central pain processing dysfunction (*Sumpton & Moulin, 2014*). Reduced physical performance capacity and levels of independence in basic activities of daily living should also be taken into account (*García-Hermoso, Saavedra & Escalante, 2015*), thus improvement in physical capacity has become an equally basic aspect of recovery.

Different studies have shown that patients suffering from FM have a reduced performance in activities of daily living (*Bravo et al., 2019*; *Da Silva Costa et al., 2017*; *Rasouli, Stensdotter & Van der Meer, 2016*), which could be due to a decrease in physical capacities such as balance and postural control (*Garrido-Ardila, González-Lopez-Arza & Jiménez-Palomares, 2020*; *Muto et al., 2015*), flexibility (*Kim et al., 2019*) and/or an increase in fatigue, a loss of strength and muscular endurance or a lower cardiac capacity (*Loftus, Dobbin & Crampton, 2021*), which could determine a decrease in physical activity and a change towards a more sedentary lifestyle (*Loftus, Dobbin & Crampton, 2021*).

One of the basic concepts pursued by rehabilitation is function and movement (*Bravo et al., 2019*). The step or gait can be considered one of the locomotor gestures with greater clinical relevance (*Heredia-Jimenez & Orantes-Gonzalez, 2019*), as it could be a very determining factor in the invidivual's independence, since coordination, semi-static and dynamic balance, muscular strength, endurance and reaction capacity in the individual's relationship with the environment are integrated in the same gesture (*Bravo et al., 2019*; *Loftus, Dobbin & Crampton, 2021*; *Natvig, Bruusgaard & Eriksen, 1998*).

The walking analysis could be performed from two different points of view: functional (*Taul-Madsen et al., 2021*) and biomechanical (*Wang et al., 2021*). The walking performance tests are aimed at evaluating the functional capacity of the invidivual in

that specific task, while the gait pattern analysis breaks down the specific gesture into biomechanical components.

Traditionally, various gait analysis methods have been used from validated tests to analyze the gait function, such as the 6MWT (*Natvig, Bruusgaard & Eriksen, 1998*; *Balke, 1963*), the Cooper KH physical running test (*Cooper, 1968*) and the Timed Up and Go test (*Podsiadlo & Richardson, 1991*), which provide standard information on the individual's ability to walk. Likewise, the implementation of some tests with instruments for kinematic measurement, such as inertial sensors (*Johnston et al., 2017*; *O'Reilly et al., 2017*), depth cameras (*Moreno et al., 2017*) and Smartphone inertial sensors (*Galán-Mercant et al., 2014*), allow analyzing the gait pattern.

In recent years, different systematic reviews have been carried out to analyze the efficacy of different types of interventions in individuals with FM to improve their physical and psychological capacities, such as aerobic exercise (*Bidonde et al., 2017*), walking (*O'Connor et al., 2015*), aquatic (*Bidonde et al., 2014*) and resistance exercise (*Busch et al., 2013*). However, to our knowledge, no systematic review has been published to date that collects and analyzes the impairment of gait, in both gait pattern and functional capacity, in the population with FM.

## Objective

The aim of the present review was to perform a systematic review and meta-analysis comparing walking test performance and gait pattern between individuals with and without FM.

## METHODS

### Bibliographic search

The present systematic review was conducted according to the guidelines and recommendations of the PRISMA statement on systematic reviews and meta-analyses (*Moher et al., 2009*). The search was performed on August 15th, 2021. This systematic review was registered in PROSPERO with the following reference: CRD42018116200.

The search for the scientific articles in this systematic review was carried out using the MEDLINE, SCOPUS, PEDRO, CINHAL and WEB OF SCIENCE databases.

The keywords used to perform the search were "Fibromyalgia", "Exercise", "Gait", "Gait Analysis", "Kinematic", "Walk", "Walking" and "Walk test". There was a constant word in all the searches, *i.e.*, "Fibromyalgia", which was combined by means of the Boolean AND/OR connectors with the rest of the conceptual groups of terms: "walk" and "gait analysis". The same search sequence was carried out in all the mentioned databases.

### Study selection

The article selection was blindly conducted by two researchers (ECV and MRM) with more than ten years of experience. Any discrepancies in the selection of documents were resolved by a third researcher (MGS).

The inclusion criteria for the selection of the studies were: (1) studies published since 2000 (to promote the reproducibility of the study); (2) studies carried out in patients

diagnosed with FM; (3) studies carried out in adult patients over 18 years of age; and (4) studies where at least one outcome variable analyzed the gait pattern and functional capacity.

The exclusion criteria were: (1) publications in a language other than English, Spanish, French, Portuguese and Italian; (2) studies carried out in patients with a BMI greater than 35 points; (3) articles in which BMI data are not mentioned; (4) studies carried out in invidivuals who had undergone surgery at least one year before the study; and (5) studies in which women were pregnant.

## Data collection process

In the same way as the selection of the articles, the data collection process was carried out by two independent blinded investigators. Data were extracted from each study about the number of participants (% women), anthropometric data (age, height, weight and BMI) and outcome variables (results on gait functional test and/or gait pattern variables: gait speed, stride length, cadence, swing, stance, single support and double support) that analyzed the gait pattern and functional capacity We extracted data from each study related to number of participants (% women), anthropometric data (age, height, weight, and BMI) and outcome variables that analyzed the gait pattern and functional capacity (results on gait functional test and/or that analyzed the gait pattern and functional capacity (results on gait functional support, and double support).

## Quality assessment

To evaluate the quality of the included studies, the Newcastle-Ottawa modified quality assessment scale (NOS) was used (*Modesti et al., 2016*). The NOS is a tool that evaluates eight aspects, distributed in three different dimensions: selection (representativeness of the sample, sample size, non-respondents, ascertainment of the exposure (risk factor)); comparability and outcome (assessment of the outcome; statistical test). Two blinded researchers carried out the evaluation of the different components of the scale.

## Synthesis of results

The obtained articles were analyzed for both gait pattern and functional capacity data on the fly in patients with FM, including any measurements from individuals without FM.

In the case of longitudinal articles, baseline measurements were used, treating the data as if the articles were cross-sectional studies.

Finally, the data were grouped according to the outcome variable and the type of test performed to obtain the measurements.

In order to provide a better estimate of the effects of FM on gait, a meta-analysis was performed on the functional capacity variable that most frequently appeared in the literature search, *i.e.*, in this case, the total meters traveled after completing the 6MWT. A total of 29 studies were integrated into the analysis. To analyze the heterogeneity of the data extracted from the included studies, the 95% prediction interval was calculated. In addition, the effect size was calculated.

The difference between the standardized means (95% CI) was calculated. The level of significance was established at $p \leq 0.05$. The I2 statistic and the chi-square test were used

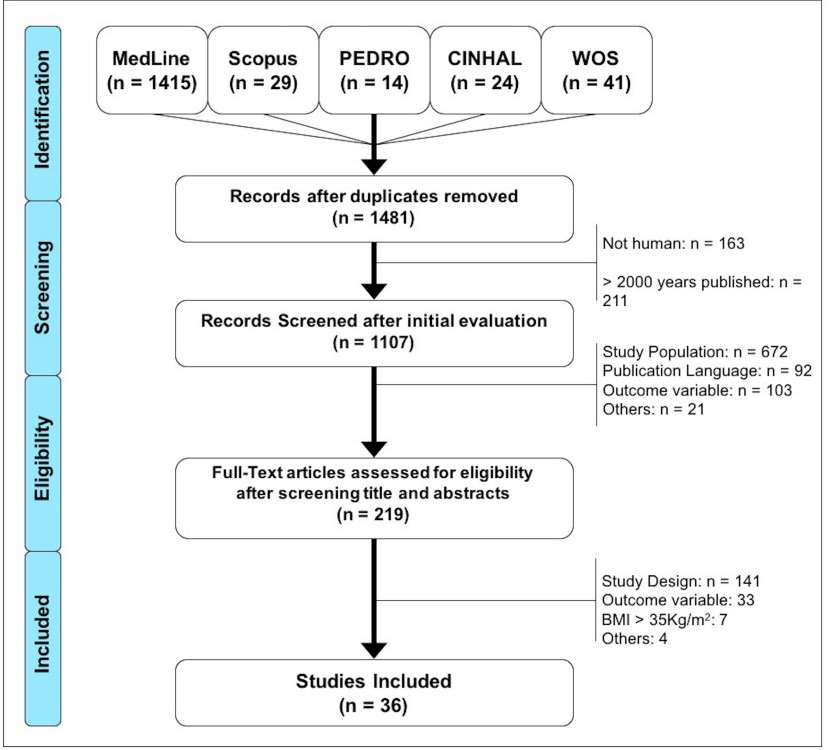

**Figure 1** **Flow diagram of the search and selection process of the documents included in this study.**

to determine the heterogeneity. The forest plot presents all studies included in the present study that analyses the 6MWT results. RevMan (Cochrane) has been the software used to perform the statistical analysis.

# RESULTS

## Search and selection of studies

After searching the different databases and eliminating duplicates, a total of 1,481 studies were found, of which 374 were excluded (163 were not conducted in humans and 211 were beyond the established publication date limit), leaving a total of 1,107 articles for screening. Subsequently, 888 were excluded after reading the title and the abstract, since their context was not in accordance with the objectives of the research. The remaining 219 articles were analyzed in full text, of which 184 were discarded. A total of 36 articles were included in the final analysis. For more details about the selection of studies, please refer to Fig. 1.

## Study characteristics

All selected documents were evaluated using the NOS scale. In this sense, the mean score was 7.9 points, with a range that oscillated between 3 and 10 (Table 1). The included studies assessed the gait pattern and functional capacity of movement through different gait analysis tests. Thirty-one studies were analyzed, with a total population of 4,078 participants (3,369 patients with FM and 709 participants without FM). The vast majority of total participants

**Table 1** Evaluation of the selected documents using the NOS scale.

| STUDY | Selection | | | | Comparability | | Outcome | | Total (x/9) |
|---|---|---|---|---|---|---|---|---|---|
| | 1 | 2 | 3 | 4 | 1a | 2b | 1 | 2 | |
| *Akkaya et al. (2012)* | * | – | – | ** | * | * | * | * | 7 (Good) |
| *Andreissy Breda et al. (2013)* | * | – | * | ** | * | * | * | * | 8 (Good) |
| *Aparicio et al. (2015)* | * | – | * | ** | * | * | * | * | 8 (Good) |
| *Auvinet et al. (2011)* | * | * | – | ** | * | * | * | * | 8 (Good) |
| *Ayán et al. (2007)* | – | – | * | ** | * | * | * | * | 7 (Good) |
| *Carbonell-Baeza et al. (2010b)* | * | – | – | ** | * | * | * | * | 6 (Fair) |
| *Carbonell-Baeza et al. (2010a)* | * | – | – | ** | * | * | * | * | 6 (Fair) |
| *Carbonell-Baeza et al. (2011a)* | * | – | – | ** | – | – | – | – | 3 (Poor) |
| *Carbonell-Baeza et al. (2011b)* | * | – | – | ** | – | – | – | – | 3 (Poor) |
| *Carbonell-Baeza et al. (2013)* | * | – | – | ** | – | – | – | – | 3 (Poor) |
| *Carbonell-Baeza et al. (2014)* | * | – | – | ** | – | – | – | – | 3 (Poor) |
| *Cardoso et al. (2011)* | * | – | * | ** | * | * | * | * | 8 (Good) |
| *Córdoba-Torrecilla et al. (2015)* | * | – | – | ** | – | – | – | * | 4 (Fair) |
| *Ericsson, Bremell & Mannerkorpi (2013)* | * | – | * | ** | * | * | * | * | 8 (Good) |
| *Ernberg et al. (2018)* | * | – | * | ** | * | * | * | * | 8 (Good) |
| *Giannotti et al. (2014)* | * | – | – | ** | * | * | * | * | 7 (Good) |
| *Góes et al. (2015)* | * | – | * | ** | * | * | * | * | 8 (Good) |
| *Heredia Jiménez et al. (2009)* | * | – | * | ** | * | * | * | * | 8 (Good) |
| *Heredia-Jimenez & Soto-Hermoso (2013)* | * | – | * | ** | * | * | * | * | 8 (Good) |
| *Heredia-Jimenez, Orantes-Gonzalez & Soto-Hermoso (2016)* | * | – | * | * | * | * | * | * | 7 (Good) |
| *Hernando-Garijo et al. (2021)* | * | * | * | ** | * | * | * | * | 9 (Good) |
| *Homann et al. (2018)* | * | – | – | ** | * | * | * | * | 7 (Good) |

*(continued on next page)*
**Table 1** (*continued*)

| STUDY | Selection | | | | Comparability | | Outcome | | Total (x/9) |
|---|---|---|---|---|---|---|---|---|---|
| | 1 | 2 | 3 | 4 | 1a | 2b | 1 | 2 | |
| *Koca et al. (2015)* | * | – | * | ** | * | * | * | * | 8 (Good) |
| *Latorre-Román, Santos-Campos & Heredia-Jimenez (2014)* | * | – | * | ** | * | * | * | * | 8 (Good) |
| *Mannerkorpi et al. (2010)* | * | * | * | ** | * | * | * | * | 9 (Good) |
| *Martín-Martínez et al. (2020)* | * | – | – | ** | – | – | * | * | 5 (Fair) |
| *Mingorance, Montoya & Miranda (2021)* | * | * | – | ** | * | * | – | * | 7 (Good) |
| *Ollevier et al. (2020)* | – | * | – | ** | – | – | * | * | 5 (Fair) |
| *Polat et al. (2021)* | * | – | * | ** | * | * | * | * | 8 (Good) |
| *Salvat et al. (2016)* | * | – | * | ** | * | * | * | * | 8 (Good) |
| *Sañudo et al. (2010)* | * | * | * | ** | * | * | * | * | 9 (Good) |
| *Soriano-Maldonado et al. (2015a)* | * | – | – | ** | * | * | * | * | 7 (Good) |
| *Soriano-Maldonado et al. (2015b)* | * | – | – | ** | – | – | – | – | 3 (Poor) |
| *Tavares et al. (2020)* | * | – | – | ** | * | * | * | * | 7 (Good) |
| *Villafaina et al. (2019)* | * | * | – | ** | * | * | * | * | 8 (Good) |
| *Vincent et al. (2016)* | * | – | * | ** | * | * | * | * | 8 (Good) |

were women (4,044 women vs 34 men). The highest mean age recorded in the individuals with FM was 56.6 (1.9) years and the lowest was 35.5 (9.9) years. Regarding individuals without FM, the highest mean was 54.1 (4.4) years, and the lowest was 38.7 (7.30) years.

The gait was analyzed both from its pattern and from the functional capacity point of view. The six-minute walk test (*Enright, 2003*) and the 8-Foot Up and Go (8UG) test (*Rikli & Jones, 1999*) were used for the functional capacity analysis. Other authors analyzed the kinematic patterns by free-walking the patient through a walkway or walkways, and with electronic devices such as motion sensors, accelerometers or depth cameras.

The population characteristics and results were grouped in 4 tables. Table 2 shows the population characteristics of the studies that analyzed the gait pattern and the tests used to obtain the data. The kinematic data and the system used for their preparation are summarized in Table 3. The following tables present the population data of the studies that analyzed the amount of walking and the results of the tests.
**Table 2** Results of the 6-minute walk test.

| Study | N [N (Women%)] | n | Age (SD) | Weight [(Kg) (SD)] | Height [(Cm) (SD)] | BMI [(Kg/m² ) (SD)] | Results (m) |
|---|---|---|---|---|---|---|---|
| Akkaya et al. (2012) | 92 (100%) | FM 51 | 35.5 (9.9) | – | – | 27.0 (4.2) | 300.9 (52.3)[*] |
| | | wFM 41 | 33.3 (7.4) | – | – | 26.3 (3.9) | 373.9 (50.5)[*] |
| Andreissy Breda et al. (2013) | 58 (100%) | FM 30 | 42.6 (5.8) | 72.4 (9.3) | 158.9 (5.3) | 28.7 (3.9) | 441.8 (84.1)[*] |
| | | wFM 28 | 40.7 (6.3) | 70.6 (13.1) | 160.7 (6.3) | 27.2 (5.1) | 523.9 (80.3)[*] |
| Aparicio et al. (2015) | 737 (100%) | FM 487 | 51.9 (8.3) | 71.3 (14.0) | 157.8 (6.0) | 28.6 (5.4) | 483.5 (89.6)[*] |
| | | wFM 250 | 49.3 (9.9) | 67.8 (12.7) | 159.8 (6.2) | 26.5 (4.6) | 586.3 (73.3)[*] |
| Ayán et al. (2007) | 29 (100%) | FM 29 | 53 (9) | – | – | 26 (4) | 432.8 (61.2) |
| Carbonell-Baeza et al. (2010a) | 65 (100%) | FM1 33 | 51.4 (7.4) | 71.2 (2.1) | – | 28.5 (0.9) | 451.9 (14.0) |
| | | FM2 32 | 50.0 (7.3) | 68.1 (2.2) | – | 27.8 (0.9) | 458.7 (15.0) |
| Carbonell-Baeza et al. (2010b) | 59 (100%) | FM1 27 | 54.2 (6.2) | 68.1 (2.2) | – | 27.5 (0.9) | 448.7 (13.5) |
| | | FM2 32 | 51.4 (7.4) | 68.5 (2.1) | – | 28.2 (0.9) | 456.6 (12.7) |
| Carbonell-Baeza et al. (2011a) | 123 (100%) | FM 123 | 51.7 (7.2) | 70.75 (13.66) | 157.26 (4.97) | 28.54 (5.60) | 447.02 (83.54) |
| Carbonell-Baeza et al. (2011b) | 6 (0%) | FM 6 | 52.3 (9.3) | 79.3 (10.7) | – | 27.1 (3.8) | 495.8 (78.0) |
| Carbonell-Baeza et al. (2013) | 118 (100%) | FM 118 | 51.9 (7.3) | – | – | 28.58 (5.6) | 438.72 |
| Carbonell-Baeza et al. (2014) | 100 (100%) | FM 100 | 50.6 (8.6) | 70.4 (12.5) | 158.6 (5.9) | 28.0 (5.4) | 473.11 (81.74) |
| Cardoso et al. (2011) | 31 (100%) | FM 16 | 53.5 (7.5) | 70.6 (14.6) | 158 (5) | 28.0 (4.9) | 446.7 (126.0)[*] |
| | | wFM 15 | 54.1 (4.4) | 70.6 (13.7) | 160 (9) | 27.5 (4.5) | 522.1 (47.6)[*] |
| Córdoba-Torrecilla et al. (2015) | 439 (100%) | FM 439 | 52.2 (8.0) | 71.3 (14.0) | 157.8 (6.0) | 28.6 (5.5) | 484.0 (81.7) |
| Ericsson, Bremell & Mannerkorpi (2013) | 133 (100%) | FM 133 | 46 (8.6) | – | – | 27.3 (5.3) | 507.0 (84.0) |
| Ernberg et al. (2018) | 255 (100%) | FM 125 | 51.2 (9.4) | – | – | 28.0 (5.2) | 551.4 (71.0)[*] |
| | | wFM 130 | 48.2 (11.4) | – | – | 24.2 (3.8) | 656.9 (60.1)[*] |
| Giannotti et al. (2014) | 32 (93,75%) | FM1 20 | 52.8 (10.6) | – | – | 24.3 (3.4) | 378 (68.4)[P] |
| | | FM2 12 | 51.3 (6.3) | – | – | 23.4 (4.2) | 403.2 (86.4)[P] |
| Hernando-Garijo et al. (2021) | 34 (100%) | FM1 17 | 51.81 (9.05) | 68.19 (16.88) | 158.63 (6.29) | 27.25 (7.30) | 403.57 (107.13) |
| | | FM2 17 | 55.06 (8.51) | 68.13 (15.10) | 161.81 (5.13) | 25.93 (5.27) | 407.01 (137.09) |
| Homann et al. (2018) | 39 (100%) | FM 20 | 39.50 (6.07) | 70.97 (14.39) | 159.38 (6.48) | 27.69 (5.43) | 473.52 (77.84)[*] |
| | | wFM 19 | 41.84 (6.18) | 72.04 (9.64) | 159.34 (5.15) | 28.40 (3.89) | 541.75 (85.62)[*] |
| Latorre-Román, Santos-Campos & Heredia-Jimenez (2014) | 50 (100%) | FM 36 | 49.8 (5.4) | 67.2 (12.1) | 157 (4) | 27.7 (4.8) | 477.5 (63.1)[*] |
| | | wFM 14 | 47.3 (5.9) | 65.6 (8.7) | 159 (6) | 25.9 (3.2) | 561.5 (73.1)[*] |

**Table 2** (*continued*)

| Study | N [N (Women%)] | n | Age (SD) | Weight [(Kg) (SD)] | Height [(Cm) (SD)] | BMI [(Kg/m²) (SD)] | Results (m) |
|---|---|---|---|---|---|---|---|
| *Mannerkorpi et al. (2010)* | 33 (100%) | FM 33 | 50 (7.6) | – | – | 28 (4.5) | 522 (56.1) |
| *Mingorance, Montoya & García Vivas Miranda (2021)* | 60 (90%) | FM1 | 50.25 (8.53) | 67.00 (7.46) | 169.15 (6.41) | 23.34 (1.23) | 391.25 (21.9) |
| | | FM2 | 52.30 (8.04) | 65.05 (5.82) | 168.25 (6.35) | 22.95 (1.30) | 365 (11) |
| | | FM3 | 54.85 (8.62) | 67.00 (7.43) | 166.90 (7.86) | 24.21 (3.93) | 385 (31.85) |
| *Ollevier et al. (2020)* | 46 (100%) | FM1 | 40.76 (6.62) | 69.76 (16.71) | 1.66 (0.06) | 25.30 (5.12) | 328.91 (98.12) |
| *Polat et al. (2021)* | 40 (100%) | FM1 20 | 47.0 (7.1) | 71.4 (10.1) | 160.0 (5.5) | 27.9 (0.80) | 443.15 (109.2) |
| | | FM2 20 | 42.6 (8.7) | 69.7 (8.03) | 162.0 (6.6) | 26.6 (0.79) | 467 (43.9) |
| *Salvat et al. (2016)* | 155 (100%) | FM1 81 | 50.0 | – | – | 26.3 | 384.0 |
| | | FM2 74 | 50.0 | – | – | 27.9 | 353.0 |
| *Sañudo et al. (2010)* | 64 (100%) | FM1 22 | 55.9 (1.6) | 72.3 (2.3) | 157 (1) | 29.6 (1.1) | 512.5 (15.9) |
| | | FM2 21 | 55.9 (1.7) | 68.5 (3.0) | 157 (2) | 27.6 (1.1) | 535.0 (16.2) |
| | | FM3 21 | 56.6 (1.9) | 74.5 (3.3) | 158 (1) | 29.7 (1.1) | 488.7 (16.9) |
| *Soriano-Maldonado et al. (2015a)* | 451 (100%) | FM 451 | 52.0 | – | – | 28.6 | 486.3 (3.71) |
| *Soriano-Maldonado et al. (2015b)* | 444 (100%) | FM 444 | 52.1 (7.9) | – | – | 28.5 (5.2) | 486 (79) |
| *Tavares et al. (2020)* | 40 (100%) | FM 20 | 40.55 (6.19) | 62.42 (7.80) | 1.60 (0.05) | 24.40 (2.58) | 546.40 (60.88)[*] |
| | | wFM 20 | 38.7 (7.30) | 63.75 (7.56) | 1.61 (0.06) | 24.70 (3.29) | 616.39 (48.05)[*] |
| *Villafaina et al. (2019)* | 37 (100%) | FM1 22 | 54.27 (9.29) | – | – | 27.11 (2.90) | 491.15 (80.21) |
| | | FM2 15 | 53.44 (9.47) | – | – | 28.19 (3.88) | 517.45 (58.42) |
| *Vincent et al. (2016)* | 60 (100%) | FM 30 | 47.0 (10.4) | – | – | 25.8 (4.6) | 570.7 (51.9)[*] |
| | | wFM 30 | 41.1 (8.4) | – | – | 26.0 (4.0) | 640.2 (65.7)[*] |

**Notes.**

FM, Fibromyalgia syndrome; P, Data obtained indirectly; BMI, Body mass index; wFM, Individuals without Fibromyalgia; FMx, Fibromyalgia subgroup.

$^{*}P < 0.05$.

**Table 3** Results of the 8 foot up and go test.

| Study | N (Women%) | n | Age (SD) | Weight (Kg) (SD) | Height (Cm) (SD) | Body mass index (Kg/m²) (SD) | Results 8UG Test (s) (Differences between groups) |
|---|---|---|---|---|---|---|---|
| *Aparicio et al. (2015)* | 737 (100%) | FM 487 | 51.9 (8.3) | 71.3 (14.0) | 157.8 (6.0) | 28.6 (5.4) | 7.02 (2.34)[*] |
| | | wFM 250 | 49.3 (9.9) | 67.8 (12.7) | 159.8 (6.2) | 26.5 (4.6) | 5.27 (1.03)[*] |
| *Carbonell-Baeza et al. (2010a)* | 65 (100%) | FM1 33 | 51.4 (7.4) | 71.2 (2.1) | – | 28.5 (0.9) | 8.5 (0.4) |
| | | FM2 32 | 50.0 (7.3) | 68.1 (2.2) | – | 27.8 (0.9) | 8.2 (0.4) |
| *Carbonell-Baeza et al. (2010b)* | 59 (100%) | FM1 27 | 54.2 (6.2) | 68.1 (2.2) | – | 27.5 (0.9) | 7.6 (0.3) |
| | | FM2 32 | 51.4 (7.4) | 68.5 (2.1) | – | 28.2 (0.9) | 8.3 (0.3) |
| *Carbonell-Baeza et al. (2011a)* | 123 (100%) | FM 123 | 51.7 (7.2) | 70.75 (13.66) | 157.26 (4.97) | 28.54 (5.60) | 8.35 (2.32) |
| *Carbonell-Baeza et al. (2011b)* | 6 (0%) | FM 6 | 52.3 (9.3) | 79.3 (10.7) | – | 27.1 (3.8) | 6.8 (1.1) |
| *Carbonell-Baeza et al. (2014)* | 100 (100%) | FM 100 | 50.6 (8.6) | 70.4 (12.5 | 158.6 (5.9) | 28.0 (5.4) | 7.31 (2.37) |

**Notes.**

FM, Fibromyalgia syndrome; wFM, Individuals without Fibromyalgia; FMx, Fibromyalgia subgroup.

[*]$P < 0.05$.

## Analysis from the walk functional capacity point of view

The most used test among the analyzed studies was the 6MWT (*Enright, 2003*), which was used in a total of 29 studies. The number of participants ranged between 6 (*Carbonell-Baeza et al., 2011b*) and 737 (*Ericsson, Bremell & Mannerkorpi, 2013*), with an exclusively female sample in all cases, except in the studies of *Carbonell-Baeza et al. (2011b)*, where the six participants were male, and *Giannotti et al. (2014)*, where 6.25% of the sample were men. The mean age of the participants ranged from 33.3 (7.4) (*Aparicio et al., 2015*) to 55.9 (1.7) years (*Sañudo et al., 2010*). The body mass index ranged between 23.4 (4.2) (*Giannotti et al., 2014*) and 29.7 (1.1) (*Sañudo et al., 2010*) in individuals with FM, while individuals without FM ranged between 24.2 (3.8) (*Ayán et al., 2007*) and 28.40 (3.89) (*Homann et al., 2018*).

Table 2 presents the characteristics and results of the studies that used the 6MWT as a walk functional capacity test.

Of all the studies included in this SR, nine of them compared individuals with FM with individuals without FM, observing significant differences in the total distance walked between both groups. For the group of individuals with FM, the shortest distance traveled in the 6MWT test was 300.9 (52.3) (*Akkaya et al., 2012*) meters, whereas for the individuals without FM group it was 373.9 (50.5) (*Akkaya et al., 2012*) meters. Similarly, the greatest distance covered for the group of individuals with FM was 570.7 (51.9) [765] meters and 656.9 (60.1) (*Ernberg et al., 2018*) meters for the individuals without FM group. The effect size observed when the values of both groups are compared (individuals with FM and individuals without FM) was ***Cohen's d:*** −1.414 (*effect-size r*: −0.578).

Other studies were included, in which the analyzed sample consisted only of individuals with FM, who, in some cases, were grouped into different intervention subgroups. As was
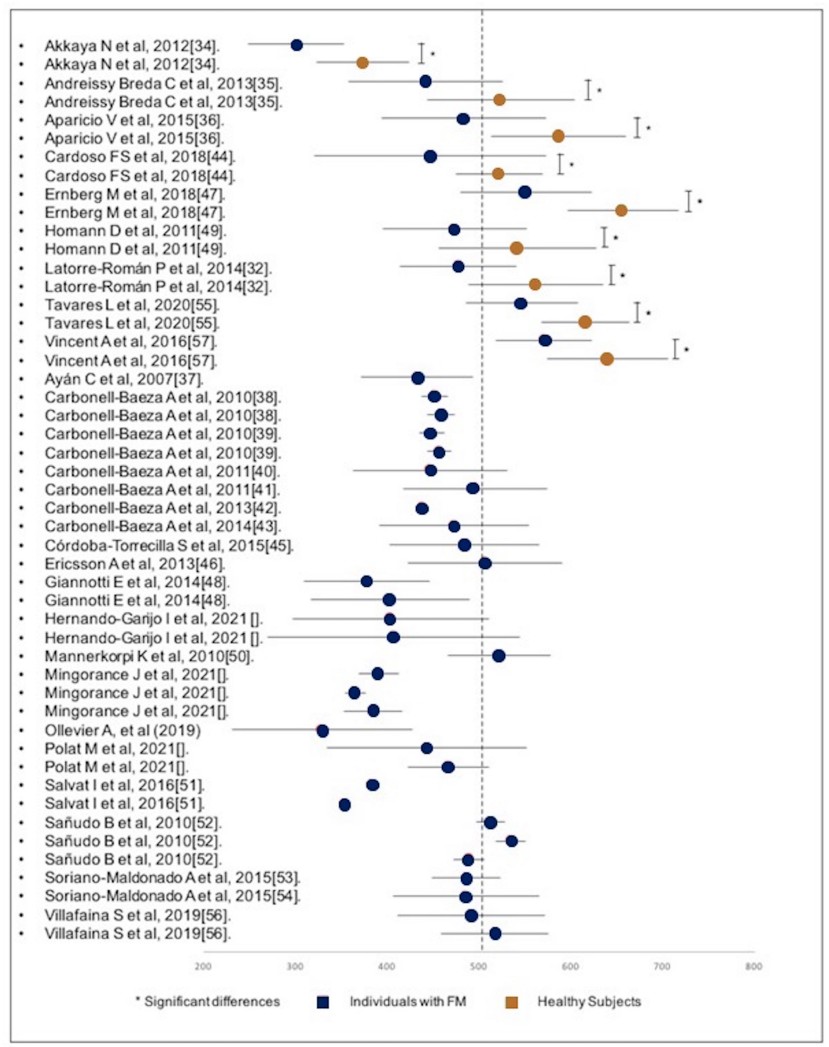

**Figure 2** Forest-plot of the distance walked during the execution of the 6MWT divided by subjects with FM and healthy subjects.

previously mentioned, only baseline measurements prior to any intervention were used. Figure 2 shows all mean values and their corresponding standard deviations.

The diagram shows that the results of the individuals without FM are greater than the majority of the results of the FM subgroups. Only the study of Akkaya (*Akkaya et al., 2012*) obtained a significantly lower result in the individuals without FM, although the result is significantly higher compared to the FM subgroup of the same study.

Another test used to analyze the walk functional capacity was the ''8 foot up and go test'' (*Rikli & Jones, 1999*), specifically in six of the analyzed studies (Table 3). Of these six studies, the only one that made a comparison between individuals without FM and individuals with FM is the study of *Aparicio et al. (2015)*, with a total sample of 737 individuals (with FM: 487/without: 250). Significant differences were obtained in the execution time of the

8UG Test (FM: 7.02 (2.34) seconds vs individuals without FM group: 5.27 (1.03) seconds) (Table 4). In a complementary way, it was observed that the best score for the FM group was 6.8 (1.1) seconds (*Carbonell-Baeza et al., 2011b*), whereas the worst score was 8.35 (2.32) seconds (*Latorre-Román, Santos-Campos & Heredia-Jimenez, 2014*). A subgroup of FM participants in the study of *Heredia-Jimenez, Orantes-Gonzalez & Soto-Hermoso (2016)* obtained the worst recorded result, with 8.5 (0.4) seconds. The data are presented in detail in Table 3.

### Analysis from a gait pattern point of view

Of the 36 studies included in this SR, 7 studies assessed the gait pattern of individuals with FM using specific tests and measurement instruments. The use of instruments such as inertial sensors has been validated in individuals with FM as an effective reference value (*Segura-Jimenez et al., 2013*). The characteristics of the gait pattern studies are summarized in Table 4.

Most of the studies obtained the kinematic data by means of a ''Walking speed test'' (authors determined a short distance to walk and then they calculated the speed) (*Góes et al., 2015*; *Heredia Jiménez et al., 2009*; *Heredia-Jimenez & Soto-Hermoso, 2013*; *Heredia-Jimenez, Orantes-Gonzalez & Soto-Hermoso, 2016*; *Koca et al., 2015*), whereas others obtained such data using the 6MWT (*Latorre-Román, Santos-Campos & Heredia-Jimenez, 2014*). In some studies, the distance was not established or described for the production of the kinematic parameters, using free walking while the individuals were analyzed. The maximum distance described for the kinematic analysis was 18.6 m (*Heredia Jiménez et al., 2009*; *Heredia Jiménez et al., 2009*; *Heredia-Jimenez & Soto-Hermoso, 2013*; *Heredia-Jimenez, Orantes-Gonzalez & Soto-Hermoso, 2016*), whereas the minimum was 10 meters (*Martín-Martínez et al., 2020*). The results of the tests are summarized in Table 4.

As can be observed in all the results, the speed (m/s) was lower in the group of individuals with FM than in the group of individuals without FM. The highest speed obtained in the group of individuals with FM was 1.94 (0.27) m/s (*Latorre-Román, Santos-Campos & Heredia-Jimenez, 2014*), compared to 2.18 (0.34) m/s (*Latorre-Román, Santos-Campos & Heredia-Jimenez, 2014*) the group of individuals without FM. The lowest speed was 0.88 (0.2) m/s (*Koca et al., 2015*) for the group of individuals with FM and 1.12 (0.3) (*Koca et al., 2015*) for the group of individuals without FM.

As can be seen, there is a significant difference in almost all the analyzed data compared to individuals without FM.

## DISCUSSION

The aim of the present study was to carry out a systematic review to verify the impairment in the gait of individuals with FM, from a gait pattern and functional capacity point of view. The kinematic variables, as well as the results of the different gait tests analyzed, show that individuals with FM have an altered gait pattern and walk less than individuals without FM.

**Table 4 Characteristics and results of studies that analyze gait pattern characteristics.**

| Study | N (Women%) | n | Age (SD) | Weight (Kg) (SD) | Height (Cm) (SD) | Body mass index (Kg/m$^2$) (SD) | Test | System | Stride Length (m) | Cadence (step/min) | Swing (%cycle) | Stance (%cycle) | Single support (%cycle) | Double support (%cycle) |
|---|---|---|---|---|---|---|---|---|---|---|---|---|---|---|
| Auvinet et al. (2011) | 104 (100%) | FM 52 | 44.1 (8.1) | – | 165 (5.8) | 24.2 (4.1) | 40 meters | Locometrix™ (three accelerometers, | – | – | – | – | – | – |
| | | wFM 52 | 44.5 (7.3) | – | 164 (6.4) | 23.8 (4.4) | | | – | – | – | – | – | – |
| Góes et al. (2015) | 32 (100%) | FM 16 | 41.5 (5.9) | – | – | 28.2 (3.9) | Walking speed | Vicon; MX-13 (6- camera Vicon motion analysis system) | 1.22 (0.09) | 58.42 (4.52) | – | | | |
| | | wFM 16 | 40.4 (6.4) | – | – | 26.8 (5.6) | | | 1.23 (0.09) (P = 0.755) | 57.41 (4.31) (P = 0.523) | – | – | – | – |
| Heredia Jiménez et al. (2009) | 99 (100%) | FM 55 | 49.5 (8.9) | 69.2 (12.9) | 157.9 (6.6) | 27.8 (5.1) | 18.6m walkway | GAITRite system; CIR Systems Inc., Clifton, NJ, USA. | 1.19 (0.14) | 109.5 (12.6) | 36.6 (2.2) | 63.4 (2.2) | 36.6 (2.2) | 26.7 (4.5) |
| | | wFM 44 | 47.1 (6.8) | 67.8 (13.4) | 157.0 (5.4) | 27.3 (5.3) | | | 1.33 (0.11). (P = 0.001) | 124.8 (8.5) (P = 0.001) | 39.0 (1.4) (P = 0.001) | 60.9 (1.4) (P = 0.001) | 39.0 (1.4) (P = 0.001) | 22.1 (2.8) (P = 0.001) |
| Heredia-Jimenez & Soto-Hermoso (2013) | 26 (0%) | FM 12 | 45.8 (7.4) | 81.1 (7.8) | 173.3 (5.2) | 27.0 (2.4) | 18.6m walkway | GAITRite system; CIR Systems Inc., Clifton, NJ, USA. | 1.37 (0.15) | 101.4 (11.8) | 36.6 (1.3) | 63.4 (1.3) | 36.6 (1.3) | 30.0 (2.5) |
| | | wFM 14 | 44.4 (7.2) | 81.9 (13.1) | 173.9 (5.5) | 27.0 (4.0) | | | 1.54 (0.11) (P = 0.002) | 112.9 (7.6) (P = 0.008) | 37.3 (1.4) (P = 0.218) | 62.6 (1.4) (P = 0.212) | 37.3 (1.4) (P = 0.218) | 25.3 (3.0) (P = 0.158) |
| Heredia-Jimenez, Orantes-Gonzalez & Soto-Hermoso (2016) | 115 (100%) | FM 65 | 49.3 (8.7) | 69.1 (11.3) | 157.1 (6.2) | 27.8 (4.1) | 18.6m walkway | GAITRite system; CIR Systems Inc., Clifton, NJ, USA. | 2.9 (1.4) | – | – | – | – | – |
| | | wFM 50 | 47.4 (6.2) | 68.7 (12.4) | 157.4 (5.9) | 27.3 (5.1) | | | 2.4 (0.9) (P = 0.04) | – | – | – | – | – |
| Koca et al. (2015) | 120 (100%) | FM 82 | 40.7 (2.0) | – | – | 26.2 (4.2) | Walking speed | Not described | – | – | – | – | – | – |
| | | wFM 38 | 38.8 (2.8) | – | – | 25.1 (2.1) | | | – | – | – | – | – | – |
| Latorre-Román, Santos-Campos & Heredia-Jimenez (2014) | 50 (100%) | FM 36 | 49.8 (5.4) | 67.2 (12.1) | 157 (4) | 27.7 (4.8) | 6MWT | GAITRite system; CIR Systems Inc., Clifton, NJ, USA | 1.36 (0.12) | 178.9 (29.8) | – | – | – | – |
| | | wFM 14 | 47.3 (5.9) | 65.6 (8.7) | 159 (6) | 25.9 (3.2) | | | 1.50 (0.14) (P = 0.001) | 201.3 (59.6) (P = 0.03) | – | – | – | – |
| Martín-Martínez et al. (2020) | 36 (100%) | FM 36 | 54.6 (10.2) | – | – | 27.29 (4.89) | 10-MWT | Functional Assessment of Biomechanics ™ (FAB) System (Biosyn System Inc.; Surrey, BC, Canada) | 1.35 (0.21) | 135.6 (21) | – | – | – | – |
| Mingorance, Montoya & García Vivas Miranda (2021) | 60 (90%) | FM1 | 50.25 (8.53) | 67.00 (7.46) | 169.15 (6.41) | 23.34 (1.23) | 6MWT Walking speed | CvMob 3.1 and CasioExilimEX-FS10 camera | – | – | 31.52 (3.44) | 61.48 (3,43) | – | – |
| | | FM2 | 52.30 (8.04) | 65.05 (5.82) | 168.25 (6.35) | 22.95 (1.30) | | | – | – | 31.77 (1.99) | 64.75 (7.24) | – | – |
| | | FM3 | 54.85 (8.62) | 67.00 (7.43) | 166.90 (7.86) | 24.21 (3.93) | | | – | – | 31.69 (2.78) | 68.30 (2.77) | – | – |

**Notes.**

FM, Fibromyalgia syndrome; wFM, Individuals without Fibromyalgia.

## Walk functional capacity analysis

One of the most widely used tests to analyze the functional capacity was the six-minute walk test, which measures the distance that a invidivual walks for 6 minutes (*Enright, 2003*). The use of the 6MWT has been recommended both in research and in the clinical setting to carry out a prior assessment or monitoring of the progression of the individual during a treatment on individuals with FM (*Mannerkorpi, Svantesson & Broberg, 2006*).

Specifically, of the 29 studies included in this SR that used 6MWT, only 9 of them made a comparison between the group of individuals with FM and individuals without FM (*Latorre-Román, Santos-Campos & Heredia-Jimenez, 2014*; *Akkaya et al., 2012*; *Andreissy Breda et al., 2013*; *Aparicio et al., 2015*; *Cardoso et al., 2011*; *Ernberg et al., 2018*; *Homann et al., 2018*; *Tavares et al., 2020*; *Vincent et al., 2016*), observing that the distance covered by individuals without FM was significantly greater than that of the FM group in all studies. In fact, it was observed that the effect size when comparing both groups was **Cohen's d:** $-1.414$ (*effect-size r*: $-0.578$). If the usual stratification on the effect size is considered ($d = 0.2$ small; $0.5$ medium; $0.8$ large (*Cohen, 1988*)), it could be stated that FM decisively affects the functional capacity of individuals with FM. Specifically, the differences observed between the the group of individuals with FM and without FM after performing the 6MWT range between 68.23 m (*Homann et al., 2018*) and 105.5 m (*Ernberg et al., 2018*). There could be several causes that justify the differences observed between the two groups. On the one hand, the gait speed in individuals with FM is altered compared to individuals without FM, as a consequence of a reduction in both the stride length and the frequency in the gait cycle (*Auvinet et al., 2006*). On the other hand, the resistance observed in individuals with FM when carrying out efforts of medium/long duration is decreased compared to individuals without FM (*Jones et al., 2008*). Therefore, a lower walking speed, added to a lower resistance in moderate efforts, could be at the base of the differences observed between the group of individuals with FM and individuals without FM. A possible explanation for the observed data can be found in the fact that the muscle recruitment pattern of individuals with FM rather resembles the fast gait pattern (poorly economical and highly energy-consuming) of individuals without FM. This pattern is determined by greater activation of the hip flexors to the detriment of the plantar flexors of the foot/ankle (*Pierrynowski, Tiidus & Galea, 2005*). However, this study identifies a gait pattern with similar characteristics between individuals with FM and without FM, which is not consistent with the results observed in the present study. In this sense, it would be interesting to design future studies that analyze whether a neuromotor re-education/retraining of individuals with FM could improve the walk functional capacity and reduce the differences with individuals without FM observed in the present study.

Taking as a reference value the distance of 554.5 m (53.29) for individuals without FM performing the 6MWT of the Enright PL study (*Enright, 2003*), the results obtained in the 29 studies (Fig. 1) show that three individuals without FM subgroups totally exceeded this mark, five individuals without FM subgroups walked distances similar to Enright's mark, and the individuals without FM subgroup of one study reached a significantly shorter distance. Regarding the FM subgroups, none exceeded this mark, and only one study shows a mean value slightly higher than this mark.

An improvement in the qualitative characteristics of the gait pattern (speed, frequency and stride length) with a previously designed protocol could improve the functional capacity outcome variables, as has already been demonstrated in other population profiles. In the study of *Kaleth, Slaven & Ang (2016)*, the administration of an exercise protocol in 187 invidivuals showed that, at the beginning of the study, the invidivuals were able to walk, in the 6MWT, 482.9 (80.5) meters, after 12 weeks of intervention they could walk 510.8 (84.4) meters, and in 36 weeks they could walk 517.7 (80.9) meters, with a mean change of 34.4 (65.2) meters ($P < 0.001$), which is very close to Enright's mark (*Enright, 2003*).

As was previously mentioned, future studies designed to improve the symptoms and functional capacities of individuals with FM should take into account the aspects presented in this systematic review.

The 8 foot up and go test has also been frequently used to quantify the degree of involvement in individuals with FM (Table 3). The results observed in this test are consistent with those obtained with the 6MWT. Once again, a walk functional capacity alteration was observed, which could be due to the alteration of the qualitative characteristics of the sample, showing in some studies that individuals with FM aged 50–60 years achieved results comparable to those observed in individuals without FM aged 80–90 years (*Rikli & Jones, 1999*). This pattern alteration is likely to be identified by different functional gait tests used in individuals with FM.

## Gait pattern analysis

Regarding the gait pattern analysis, Table 4 shows lower results in all the comparisons of the kinematic variables for the individuals with FM compared to the individuals without FM. When comparing the results obtained in the gait pattern analysis between individuals with FM and individuals without FM, it is observed that individuals with FM moved significantly more slowly, with a stride length and cadence lower than that of individuals without FM (Table 4). However, in other characteristics that are used for the gait pattern, such as the percentage of time used in monopodal and bipodal support or rocking, they do not present significant differences (Table 4). A possible explanation for these results can be found in the fact that individuals with FM have less mobility in the subtalar joint, which affects the plantar/dorsal flexion of the foot, increasing the dynamic imbalance of the invidivual and the risk of falls in these invidivuals (*Góes et al., 2015*; *Silva et al., 2016*). This indicates that individuals with FM, before this self-perceived instability, preventively reduce the length and cadence of their gait in order to maintain the control of the movement. This reduction in the biomechanical characteristics of gait is similar to that observed in the elderly (*Rikli & Jones, 1999*). *Góes et al. (2014)* showed that middle-aged women with FM have a gait pattern similar to that of older women, characterized by reduced ROM of the lower extremities, stride length and speed gait, thus it could be affirmed that individuals with FM suffer an adaptation in biomechanics similar to that which occurs in aging (*Rikli & Jones, 1999*).

Moreover, individuals with FM could be conditioned by psychological factors that could be negatively affecting the cortical and sub-cortical mechanisms of gait control,

by sharing the cognitive functions involved in chronic pain (*Auvinet & Chaleil, 2012*). The most obvious result of this conditioned perception of pain is fear of movement or kinesiophobia, which has been correlated in previous studies with an alteration in normal functional movement patterns (*Shigetoh et al., 2019*). Despite the fact that a SR published in 2012 associated, in individuals with FM, a perception of fatigue during the execution of the 6MWT with a reduction in the general state of health measured with SF-12 (*Auvinet & Chaleil, 2012*), other studies have correlated kinesiophobia with walk functional capacity analysis in individuals with FM. The importance of cognitive impairment in walking is reinforced in the study of Lundin-Olsson L (*Lundin-Olsson, Nyberg & Gustafson, 1997*), who analyzed how frail or cognitively disabled older people could not walk while performing a secondary task such as speaking, which is in line with the results obtained in invidivuals with similar involvement (*Eyskens et al., 2015*).

In parallel, a walking speed of ≤0.8 m/s has been associated with "low physical performance" (*Roll et al., 2003*; *Laurentani et al., 2003*). However, despite the fact that a decrease in gait pattern has been observed in individuals with FM, only the study published by *Koca et al. (2015)* presents a walking speed close to the indicated threshold. It may be necessary to redefine the concept of "low physical performance", since all the studies analyzed in the present SR show a biomechanical deficit during ambulation. The value of 0.8 m/s has been used as a predictive factor of mortality in different population groups (*Kamiya et al., 2017*; *Roshanravan et al., 2013*; *Studenski, 2011*), although, based on the results observed in this SR, "low physical performances" can be identified in values above the aforementioned threshold. Nevertheless, it is important to clarify that this reference value is preferably used in older people, thus there is rather a decrease of functional capacity in individuals with FM, with a decrease in functional capacities, with respect to normality, being frequently associated with problems of different types.

Only one study compares the gait pattern in similar population groups of different sex, *i.e.*, the study of *Heredia-Jimenez & Soto-Hermoso (2013)* which was conducted in men only ($N = 26$); the results of men with FM ($n = 12$) were slightly higher than the rest of the data obtained from the female population, with a speed of 1.16 (0.20) m/s for males with FM, which was also higher than that of the male without FM population, with 1.45 (0.19) m/s. In this sense, it would be necessary to design new studies to identify eventual differences in the gait pattern and/or walk functional capacity between sexes, so that they are taken into account when assessing and monitoring individuals with FM.

Other studies have shown a decrease in functional capacities and an alteration in the correct execution of the gait pattern when individuals are forced to perform cognitive tasks during the execution of the gait (*Yogev-Seligmann et al., 2013*; *Hagner-Derengowska et al., 2016*). The literature has identified that individuals with FM have a specific cognitive problem that could be at the base of a worse functional performance (*Bell et al., 2018*). It would be interesting in the future to design studies comparing basic functional abilities, such as walking, between individuals with FM and individuals without FM carrying out dual tasks during the different tests, to analyze whether the results are comparable between both studies. In the same way, the affectation suffered by individuals with FM could

be analyzed when they perform functional tasks such as walking, while simultaneously performing cognitive tasks.

Furthermore, it is important to bear in mind that different studies have identified deficits in static and dynamic balance in individuals with FM (*Núñez Fuentes et al., 2021*; *Jones et al., 2011*), and this imbalance could be associated with a greater risk of falling (*Núñez Fuentes et al., 2021*; *Collado-Mateo et al., 2015*). These aspects could justify the alteration in the gait pattern that causes the individuals with FM to perform a more conservative ambulation to minimize the risk of falling.

In addition, on those documents that include invidivuals with a BMI greater than 30 kg/m$^2$ but less than 35 (exclusion criterion), it should be considered that some alterations identified in the gait may be due to the BMI, thus it would be advisable to interpret these results cautiously.

### Strengths and weaknesses

Although this study is the first to analyze and integrate the gait pattern with functional gait abilities in individuals with FM, some weaknesses have been identified that should be taken into account when interpreting the results. Although a bibliographic search was carried out in 5 different databases, there could be relevant documents published in other databases. In the same way, although the documents published in 5 languages were selected, there could be documents published in other languages that were not included in the present study. Finally, it is important to consider that the score of the various selected studies, using the NOS scale, present a medium-high valuation, with few studies obtaining the maximum score and some studies showing a poor score.

# CONCLUSIONS

Compared to individuals without FM, individuals with FM demonstrate lower functional performance, observing that they walk significantly less during the 6MWT. In addition, the gait is performed with an altered gait pattern, so there could be a relationship between the altered gait pattern and lower functional efficiency, thus it would be necessary to investigate different non-pharmacological interventions in individuals with FM. (as for example neuromotor re-education, between others) would imply a recovery of the walk functional capacity, reaching the levels of individuals without FM.

### Funding
The authors received no funding for this work.

### Competing Interests
The authors declare there are no competing interests.

### Author Contributions
- Elio Carrasco-Vega, María Ruiz-Muñoz, Antonio Cuesta-Vargas, Rita Pilar Romero-Galisteo and Manuel González-Sánchez conceived and designed the experiments,

performed the experiments, analyzed the data, prepared figures and/or tables, authored or reviewed drafts of the paper, and approved the final draft.

## Data Availability

This study is a Systematic Review.

## Supplemental Information

Supplemental information for this article can be found online at http://dx.doi.org/10.7717/peerj.12908#supplemental-information.

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
