# Peer review of "Individuals with fibromyalgia have a different gait pattern and a reduced walk functional capacity: a systematic review with meta-analysis"

_PeerJ, doi:10.7717/peerj.12908_

## Round 0.1 · original submission · Major Revisions

Dear Authors

Three reviewers have conducted an in-depth analysis of your manuscript, all three agree that the systematic review has a lot of scientific value but there are major changes that need to be made, since in the current state it cannot be published.

I agree very much with the reviewers' suggestions and consider that all of them should be addressed. The following are the most important issues that should be addressed:

1. It is necessary to include an evaluation of methodological quality, the NOS scale could be an appropriate instrument for its review.

2. Better define the objectives.

3. The search strategy must be specifically included by database to comply with the principle of reproducibility.

4. I believe that the conclusions of this study do not adjust to the many limitations of the findings obtained.

·

Basic reporting

- First of all, I think there are some formal flaws in the wording of the manuscript. Some sentences should be clearer. I recommend authors to request a language editing service to improve these aspects.

- The beginning of the introduction is adequate, but I believe that it does not sufficiently address the knowledge and relevance of this topic. The main point of this manuscript is gait in patients with FM, but they only mention this aspect superficially in lines 69-74. Furthermore, they refer to an opinion editorial (reference 14) on central sensitization syndromes that is not directly related to the main interest of this review.

- In addition, it would be interesting to comment on previous literature in this field.

- It is necessary to improve the resolution of Figure 1, since it is not possible to see the content in detail.

Experimental design

- Objective: Should be better written (to assess the impact of fibromyalgia on FM patients?).

- It would be interesting to describe in detail the search strategy (line 100)

- Why was the year 2000 used as a selection criterion? Similarly, why was the BMI cut-off point set at 35 and not others?

- An important methodological limitation of this review is the lack of risk of bias assessment of the included articles. I suggest that the authors use some appropriate scale for this purpose, such as the Newcastle-Ottawa modified quality assessment scale (NOS). In the same way, the certainty or quality of the evidence found should be evaluated.

- Throughout the manuscript, and specifically in lines 134-137, it is described that a meta-analysis of the 6MWT variable was performed. However, it is not described how the statistical analysis was performed. Was a statistical aggregation of the data actually performed or were they only collected? I do not know if this should be called meta-analysis.

- Another of my main concerns is that only 9 compared the outcomes of FM patients with healthy subjects. The main objective of this review is to quantitatively and qualitatively assess the gait of FM patients, but this is not possible without adequate comparison groups.

Validity of the findings

- The validity of the findings is totally influenced by the methodological limitations of this review.

- I do not think it is possible to draw general conclusions from the results obtained.

Additional comments

In general terms, I think the article can be interesting and add some value in this field of knowledge.

However, I suggest to the authors to totally revise and improve the methodology used. It is necessary to perform meta-analyses that really evaluate the presence of gait disturbances in these patients, to consider whether it is relevant to include studies that do not present a control group and to assess the methodological quality of the included studies.

·

Basic reporting

- This manuscript has clear and unambiguous, professional English used throughout.
- The literature references, sufficient field background/context are provided.
- It presents Professional article structure, figures, tables. However, the figures present poor quality (very hard to read). Finally, the raw data was not shared.
- Self-contained with relevant results to hypotheses. However, there are aspects in the methods that should be improved.

Experimental design

- The manuscript presents original research within the aims and scope of this journal.
- The research question is well defined, relevant & meaningful. This manuscript is clear on how the research fills an identified knowledge gap.
- A rigorous investigation is performed to a high technical & ethical standard. However, there are aspects in the methods that should be improved.
- The methods are described in detail, but there is not enough information to replicate at the current state.

Validity of the findings

- The rationale & benefit to literature is well written. However, some concepts and definitions should be revised.
- The analysis needs more clarification in order to be robust, statistically sound, & controlled.
- Conclusions need more clarification while linking to the research question and supporting the results.
- The discussion is well based on the literature and brings explanations related to mechanisms that could explain some of the findings.

Additional comments

1. Thank you for allowing me to review this interesting and timely work. It is a very interesting systematic review about fibromyalgia and altered gait and walking performance. However, there are a few points that the authors should address. Please take a look at my specific comments below (attached).
2. The state aim "was to verify qualitatively and quantitatively the impact of fibromyalgia on the gait pattern of patients with FM". However, the objective state here does not align with the further described statistical analysis. My understanding (after reviewing this SR) is that the goal "was to perform a systematic review and meta-analyses comparing walking test performance and gait pattern between individuals with and without fibromyalgia”.
3. I suggest the search be more cohesive by having a search strategy focused on combining three conceptual groups of terms: (1) fibromyalgia; (2) walk (performance) tests; and (3) gait analysis.
4. I also suggest clarification about the difference between walking test performance and gait pattern analysis.

·

Basic reporting

in discussion I would suggest to deepen the part between:
- speed and fatigue
- cognitive function and gait cycle
If possible, in the results part, also reporting in a separate table, if in the various articles analyzed specific data and scales are reported to measure fatigue and cognitive function.

Experimental design

no comments

Validity of the findings

I suggest to report, please, the quality level of the papers analyzed according to the scales most convenient for you (for example PEDRO or others)

Additional comments

Dear Authors,

please,
in discussion section I would suggest to deepen the part between:
- speed and fatigue
- cognitive function and gait cycle
If possible, in the results part, also reporting in a separate table, if in the various articles analyzed specific data and scales are reported to measure fatigue and cognitive function.

Also,
I suggest to report, the quality level of the papers analyzed according to the scales most convenient for you (for example PEDRO or others) (in Result Section)

Also,
It would be important to suggest or hypothesize the importance of your results in the light of the rehabilitation treatment or adapted physical activity (by inserting specific exercises or paths on this topic) ( at the of discussion and conclusion).

There are some studies that report a decreased proprioceptive deficit in FM patients: so, if we feel more unstable, maybe we go slower because we are afraid of falling? or to move?
if the articles you have analyzed give you this possibility, the discussion could also be enriched with these aspects

---

## Round 0.2 · Minor Revisions

Dear Authors

I congratulate you for the effort you have made in modifying the manuscript, I consider that your systematic review has improved it very significantly.
The reviewers' comments are very positive but there are still some issues that need to be addressed before the paper can be finally accepted.
I recommend that the authors respond to the issues raised by the reviewers. It is important that further explanation of how the meta-analysis was developed is addressed in depth.

·

Basic reporting

The authors have significantly improved their manuscript based on the above comments. It would be interesting to include more information about how the meta-analysis was performed in the methods section.

Experimental design

Ok

Validity of the findings

Ok

·

Basic reporting

• Clear, unambiguous, professional English language used throughout?
o Please review some paragraphs where the idea is a bit confusing (notes/comments are provided in the main track changes document)
o Please be consistent with the use of acronyms e.g. using FM constantly over the text instead of fibromyalgia (or vice-versa). My suggestion is to pick one.
o Please substitute the word “subjects’ with “individuals”, “people”, or “persons” throughout your text.
o Please substitute the word “healthy” with “without FM” or “control” throughout your text.
• Literature references, sufficient field background/context provided?
o The article includes sufficient introduction and background to demonstrate how the work fits into the broader field of knowledge. Relevant prior literature is appropriately referenced. (specific notes/comments are provided in the main track changes document)
• Professional article structure, figures, tables. Raw data shared?
o The structure of the article is in conformity to the ‘standard sections’.
o Most of the figures are relevant to the content of the article: Figure 2 would e more relevant with the meta-analysis results.
o Figures have now good resolution and are appropriately described and labelled.
o Tables 1: Please add information on the meaning of the numbers in the table. On the table heading, please substitute the numbers with a short expression of what is being assessed through NOS.
o It would be interesting to have a table with the excluded studies (as supplement material).
• Self-contained with relevant results to hypotheses?
o The submission is ‘self-contained,’ represents an appropriate ‘unit of the publication’; however, it does not include all results relevant to the hypothesis. The process of how the meta-analysis was carried out is not clear. It misses important information to be able to affirm or not the hypothesis.
- Important points to be clarified: how did you perform the meta-analysis? It is missing the description of how this statistical analysis was performed. For example: “Did you used Hedges’ g to calculate the standardized mean difference (SMD) between individuals with and without FM using the means, standard deviations (SD) and sample sizes of both groups? If the full text did not contain SD, did you calculate SD from the reported standard error or confidence intervals (CI)? Did you use random-effects models with an inverse variance (IV) method to pool the SMD of included studies? Did you calculate the heterogeneity with I-squared (I2)? Did you estimate publication bias visually via re-displayed funnel plots? Did you use Duval and Tweedie’s Trim and Fill test to estimate the adjusted (or unbiased) pooled SMD (if the case)?

Experimental design

• Original primary research within the scope of the journal?
o Yes
• Research question well defined, relevant & meaningful. It is stated how the research fills an identified knowledge gap?
o The submission clearly defines the research question, which is very relevant and meaningful to the area. The knowledge gap being investigated is identified, and statements are made as to how the study contributes to filling that gap. However, I suggest including in the text the reason for not including articles before 2000 (for transparency) since the FM diagnosis was established in 1990 and possibly performance tests as the 6MWT and 8fUP could have been used in studies before 2000. An important aspect that should be included is the test Time up and go: why this test was not included? It assesses the same parameters as the 8fUP (the only difference is the distance to the cone).
• Rigorous investigation performed to a high technical & ethical standard?
o The investigation can improve the statistical analysis in order to be conducted rigorously and to a high technical standard. The research was conducted in conformity with the prevailing ethical standards in the field.
• Methods described with sufficient detail & information to replicate?
o Methods are Not yet described with sufficient information to be reproducible by another investigator. Details on how the Meta-analysis was performed need to be presented: how did you perform the meta-analysis? The article is missing the description of how this statistical analysis was performed. I also suggest that authors be more careful with the language used while interpreting study findings based on qualitative analysis of studies not included in the meta-analysis (when this was not possible). More details are provided in the main track changes document).

Validity of the findings

• Impact and novelty not assessed?
o The impact, degree of advance, and novelty of this systematic review are of interest to multiple audiences. More information related to the meta-analysis must be provided for the study replication, and to add more value to the literature.

• All underlying data have been provided; they are robust, statistically sound, & controlled?
o The data on which the conclusions are based still misses information related to the meta-analysis to be robust, statistically sound, and controlled. I also suggest that authors be more careful with the language used while interpreting study findings based on qualitative analysis of studies not included in the meta-analysis (when this was not possible). More details are provided in the main track changes document).
• Conclusions are well stated, linked to original research question & limited to supporting results.
o The conclusions still need work to be better connected to the original question investigated, and should be limited to those supported by the results. The conclusions are based still misses information related to the meta-analysis

Additional comments

Additional comments are provided in the "peerj-60354-Main_Document_R1_PRS_Reviwer 3"

·

Basic reporting

no comment

Experimental design

no comment

Validity of the findings

no comment

Additional comments

no comment

---

## Round 0.3 · accepted · Accept

Dear Authors

I congratulate you for all the improvements you have made to the manuscript, the study has a good methodological quality and the findings may have clinical utility, for this reason my final decision is to accept the study.

·

Basic reporting

- Literature references are provided and a sufficient field background/context is present in the text.
- Good quality figures and tables
- Relevant results to hypotheses.

Experimental design

- This review is original research within the aims and scope of the journal.
- Research question is well defined, relevant & meaningful.
- Methods described with sufficient detail & information to replicate.
A few suggestions for improvement:
- Review the section "Data Collection process": there is repetitive information between lines 137 to 146.
- Line 163: I suggest the use of the word "impact" instead of "effect" of the FM.

Validity of the findings

- In the discussion, it's needed less information already present in the results section and focus more on the "how" they add value to the literature
- In the conclusion, I suggest including information related to the quality of the studies from which the conclusions were drawn.